# Perspectives on Vascular Regulation of Mechanisms Controlling Selective Immune Cell Function in the Tumor Immune Response

**DOI:** 10.3390/ijms23042313

**Published:** 2022-02-19

**Authors:** Michael Welsh

**Affiliations:** Department of Medical Cell Biology, Uppsala University, Husargatan 3, Box 571, SE-75123 Uppsala, Sweden; michael.welsh@mcb.uu.se; Tel.: +46-184-714-447

**Keywords:** vasculature, immune cells, tumors, cytokines, cell adhesion molecules, leakage

## Abstract

The vasculature plays a major role in regulating the tumor immune cell response although the underlying mechanisms explaining such effects remain poorly understood. This review discusses current knowledge on known vascular functions with a viewpoint on how they may yield distinct immune responses. The vasculature might directly influence selective immune cell infiltration into tumors by its cell surface expression of cell adhesion molecules, expression of cytokines, cell junction properties, focal adhesions, cytoskeleton and functional capacity. This will alter the tumor microenvironment and unleash a plethora of responses that will influence the tumor’s immune status. Despite our current knowledge of numerous mechanisms operating, the field is underexplored in that few functions providing a high degree of specificity have yet been provided in relation to the enormous divergence of responses apparent in human cancers. Further exploration of this field is much warranted.

## 1. Introduction

Recent advances have highlighted the importance of the immune response in tumor expansion and metastasis. In a simplistic manner, an immune response can be described as pro-tumoral or anti-tumoral, of which the first category primarily reflects type 2 macrophages [1], myeloid-derived suppressor cells (MDSC) [2] and T and B regulatory cells (Tregs and Bregs) [3] and the second category type 1 macrophages, T and B effector cells, natural killer (NK) cells and dendritic cells (DC) [4,5,6,7]. Thus, there might be pro-tumoral and anti-tumoral responses operating simultaneously and the net effect will depend on the relative strengths of the antagonistic responses. Whereas the tumor itself may directly regulate the recruitment and activities of relevant—acquired and innate—immune cell (IC) populations, one can envisage that a similar control of the immune response may be exerted by the vasculature, since many IC are recruited from the blood and thus require active extravasation to infiltrate the tumor.

For IC extravasation, the leukocyte is required to cross the vascular barrier, which consists of endothelial cells (EC), pericytes and a basement membrane [8,9]. This occurs primarily at venular sites [8,9]. The IC may passage the EC layer either via paracellular migration at junctions or by directly crossing the EC by transcytosis. EC junctions consist of adherens (AJ) and tight (TJ) junctions of which the former have been extensively studied whereas much less is known about the latter. Leakage of plasma proteins, for which the process “vascular permeability” is a prerequisite, primarily occurs by disassembly of AJ [9]. AJ are generated by the assembly of homophilic dimers *in trans* of the cell surface protein vascular endothelial cadherin (VE-cadherin) [10]. Other proteins involved in IC extravasation are junctional adhesion molecules (JAM) JAMA, JAMB, JAMC, CD31/platelet endothelial cell adhesion molecule (PECAM1), ESAM (endothelial cell-selective adhesion molecule), CD99 and CD99L2 [10]. AJ are also attachment sites for the cytoskeleton via alpha (α)- and beta (β)-catenin [10] and interplay with focal adhesions via focal adhesion kinase (FAK) [11]. AJ disassembly is a complicated process occurring in response to inflammatory stimuli involving phosphorylation events primarily on VE-cadherin, cytoskeletal retraction and VE-cadherin endocytosis. The end result is an “open” AJ allowing passage of plasma macromolecules. Despite this open state, parts of the junctions remain intact, preventing EC–EC detachment/dissociation under most conditions. This separates leakage from leukocyte transmigration since the latter cannot occur if there are remaining physical obstructions. Little is known on how TJ are disassembled when leakage is increased. In addition, pericytes and the basement membrane form additional barriers that must be disrupted for leakage to occur [12,13,14]. In addition to vascular barrier properties, cytokine/chemokine-derived cues present in the microenvironment operate on the leukocytes, stimulating their mobility and propensity to transmigrate [10]. This review will explore processes by which the vascular barrier controls tumor IC infiltration. 

The abnormalities of the tumor vasculature in relation to the immune cell response have been extensively reviewed [15,16] and there is a significant amount of knowledge demonstrating mechanisms of leukocyte endothelial transmigration. However, less is known about selective mechanisms by which the vascular barrier may promote/restrict access of selective immune cell populations to the tumor, thus serving a gate-keeper function. The present review will briefly summarize characteristics of the tumor vasculature and IC extravasation. The main focus will be on selective mechanisms shifting the IC pro-tumoral/anti-tumoral balance. The complexity of the EC/IC interdependence allows for a plethora of responses that have the potential for intervention therapy.

## 2. Leukocyte Extravasation

Several reviews have excellently described the process of leukocyte extravasation, so presently that process will only briefly be discussed by summarizing key features described in detail elsewhere [10,17,18]. Extravasation is a multi-step process that begins with leukocyte capture at the EC surface, followed by rolling, arrest, crawling and transmigration of the cell. Although transmigration through the EC layer may occur through the EC itself (transcytosis), the most favored path is by paracellular migration at junctions. It has become clear that this process does not normally result in parallel leakage of plasma proteins and thus commonly is distinct from leakage as described above.

The early stages of capture and rolling involve transient interactions of P- and E-selectin on the EC surface with P-selectin glycoprotein ligand 1 on the leukocyte. This is followed by leukocyte arrest and crawling, which require the intercellular adhesion molecules (ICAM) ICAM-1, ICAM-2 and vascular cell adhesion molecule-1 (VCAM-1) binding to leukocyte lymphocyte function-associated antigen 1 (LFA1), macrophage antigen 1 (MAC1) and beta (β) 1 integrin very late antigen 4 (VLA4). During this process, the leukocyte scans for possible transmigratory sites which normally will be paracellular but may be transcellular. Once a paracellular exit site has been located, i.e., at an EC junction, a complicated reaction ensues involving clustering of ICAM-1, dissociation of junctions, cytoskeletal retraction, VE-cadherin phosphorylation events, protease activity and EC endocytosis of junctional components. EC junctions will seal up around the leukocyte during and subsequent to its diapedesis by creating dome-like structures around and behind the leukocyte [19], preventing leakage of plasma proteins. Different proteins will play distinct roles during this process. JAMA will be located on the apical EC side guiding the leukocyte at an early stage to the transmigratory site. ESAM, JAMB and JAMC will support further diapedesis through the gap opening between the ECs while VE-cadherin is simultaneously dispersed. Exit over the basement membrane requires CD99, CD99L2 and CD31/PECAM1, and JAMC prevents reverse transmigration of the leukocyte. In addition, the leukocyte must pass the pericyte and basement membrane barriers but not much is known about these processes. The complexity of vascular barrier transmigration allows for selective entry of leukocytes under various conditions, thus suggesting a role for EC to partake in the control of tumor IC infiltration. 

## 3. EC Cell Adhesion Molecules, Junctions and Leukocyte Extravasation in Tumor Biology

### 3.1. General Properties of Tumor Angiogenesis and Vasculature

The tumor vasculature commonly exhibits a number of vascular abnormalities with functional consequences resulting from chronic hypoxia and vascular endothelial growth factor-A (VEGFA) overproduction, causing protracted angiogenic stimulation [20]. These include irregular vessel parameters (size, tortuosity, branching, vessels without connections), discontinuous endothelial lining, a low degree of pericyte coverage, poor perfusion, increased leakage and high interstitial pressure. Based on these characteristics, the concept “tumor endothelial barrier” has been presented [21]. Anti-angiogenic treatment will reverse many of these vascular abnormalities and that process has been given the name “vascular normalization” [22]. The tumoral vascular phenotype has multiple consequences for leukocyte extravasation. On the one hand, there is hypoxia and metabolite accumulation, conditions that may have adverse effects on leukocyte recruitment and function. In addition, the poor blood flow and perfusion may limit leukocyte access to the tumor. On the other hand, decreased endothelial and pericyte lining with larger gaps between the cells and open junctions may increase IC access to the extravascular tumoral space. Finally, the EC display numerous alterations in their properties due to excess, chronic VEGFA stimulation that could have an impact on the ability of leukocytes to extravasate. These include changes in cell surface expression of adhesion and junctional proteins, properties of the cytoskeleton, configuration and activities of junctional proteins, endocytotic processes, phosphorylation events and other post-translational modifications. 

### 3.2. VEGFA/VEGFR2 Signaling

Since VEGFA is a major player in tumor endothelial biology, a brief summary of VEGFA-dependent EC events will be provided. Activation of VEGFR2 on the EC will start signaling that stimulates leakage, EC proliferation and angiogenesis. Important signaling pathways are activation of phospholipase C gamma, extracellular-regulated kinase (ERK), phosphatidyl inositol 3′ kinase (PI3K), FAK, Src-family kinases (SFK) and Rho-family GTPases that will break AJ by VE-cadherin phosphorylation, cytoskeletal retraction and endocytosis [23]. The adapter proteins TSAd (T cell-specific adapter protein) and SHB (Src homology-2 domain protein B, coded for by the *Shb*-gene) significantly contribute to conveying these responses, TSAd by recruiting SFK to VEGFR2 [24] and SHB by linking VEGFR2 to FAK activation [25,26,27]. These responses will facilitate disassembly of the endothelial barrier and IC extravasation by TSAd/SFK and FAK, causing VE-cadherin tyrosine phosphorylation [24] and AJ disassembly [11]. However, it should be noted that leukocyte extravasation may occur without concomitant vascular leakage [10]. In addition to immediate signaling events that change EC properties, VEGFA may reprogram the EC gene expression profile to confer additional structural changes to the EC cytoskeleton, intercellular junctions, cell surface adhesion receptor expression and cytokine/chemokine expression.

### 3.3. Consequences of Alterations in EC Cell Adhesion and Junction Protein Expression/Activities

The angiogenic environment of the tumor has a propensity to reduce IC adhesion to EC, a functional state commonly referred to as EC anergy [28,29]. Changes in tumor EC properties under inflammatory conditions with direct consequences for leukocyte tumor infiltration are altered expression of P- and E-selectin [30], ICAM-1 and ICAM-2 [31], VCAM-1 [32] and CD31/PECAM-1 [33]. E-selectin is relatively important for neutrophil and memory T cell extravasation [34], ICAM-1 and VCAM-1 for T lymphocyte and monocyte movement into inflamed tissues [32] whereas ESAM depletion selectively reduces neutrophil transmigration [35]. ICAM-1 and VCAM-1 have been found to be deregulated in tumor EC [21], possibly due to the high expression of VEGFA [31,36]. Concerning the JAMs (JAMA, JAMB, and JAMC), alteration of JAMA and JAMC activities has been found to selectively perturb leukocyte tissue infiltration under various inflammatory conditions [37]. Since JAMA and JAMC are expressed on both EC and leukocytes, it is difficult to assess whether the reported effects relate to conditions intrinsic to EC, IC or both [37]; but in one study with JAMA deficiency in EC [38], neutrophil extravasation was impaired, whereas in another study exploring forced overexpression of JAMC in EC, reduced leukocyte extravasation was noted [39]. SFK-dependent phosphorylation of JAMA has also been reported [40]. In contrast, the expression of EC cell surface adhesion molecules required for IC extravasation is commonly upregulated in tumor EC following anti-angiogenic treatment [41], allowing facilitated entrance of IC [42]. 

Increased VE-cadherin expression in tumor EC will increase tumor T cell infiltration [43], specifically demonstrating the importance of AJ for IC extravasation. VE-cadherin has been shown to be phosphorylated at residues Y658, Y685 and Y731 [9,10]. Whereas dephosphorylation of Y731 is required for leukocyte extravasation [10], Y685 phosphorylation is required for cytokine-induced vascular leakage and has possible relevance for IC transmigration [9]. Y685 phosphorylation is necessary for triggering VE-cadherin endocytosis and loosening of AJ [9].

CD31/PECAM-1 is a target of tyrosine phosphorylation by SFK and this provides possible means for regulation of IC extravasation [44]. CD31/PECAM-1 also binds the cytoskeleton, which probably participates in cytoskeletal changes required for junction remodeling. CD31/PECAM-1 phosphorylation is increased by interactions with extracellular matrix proteins, mechanical stress, shear stress and osmotic shock and decreases in migrating EC [44]. Evidence suggests that CD31/PECAM-1 phosphorylation operates as an inhibitor of receptor tyrosine kinase signaling in EC, modulating the EC response [44]. The findings above have been summarized in Table 1.

Monocytic/macrophagic MDSC infiltration into breast carcinoma tumors was increased in mice exhibiting *Shb* deficiency in EC [45], an effect that correlated with increased metastasis. In that study, an EC gatekeeping function was likely to be altered in such a manner that an IC population promoting recruitment and expansion of MDSC was specifically allowed to transmigrate over the vascular barrier. *Shb*-deficient EC exhibit altered signaling properties as indicated above with consequences for leakage [27,46], IC recruitment [47], junction morphology [46,48] and gene expression profiles pertaining to the cellular gene ontology components of focal adhesions and AJ [48].

### 3.4. Pericytes and IC Infiltration

In addition to the endothelial barrier, pericytes may control IC vascular transmigration into tumors [9,13,49]. The effects may not only relate to direct barrier properties but may also be secondary to effects of pericytes on EC [50] or tumor cells [51]. Tumor blood vessels exhibit pericyte abnormalities [52]. Melanomas grown in mice with *Shb*-deficient pericytes exhibit decreased pericyte coverage, increased leakage and increased metastasis [48]. No difference in IC infiltration was detected in that study. However, in another study, pericyte deficiency increased vascular transmigration of myeloid-derived suppressor cells (MDSC) [53,54]. 

**Table 1 ijms-23-02313-t001:** Summary of findings described in the text in relation to EC expression of cell surface adhesion molecules, local cytokine/chemokine conditions, consequences on IC extravasation and relevant references.

EC Cell Surface Adhesion Molecule/Junction Protein	Condition that Causes Alteration	Consequence	References
P-/E-selectin	TNF-α, IL-1β, IL-4, IFN-γ	Selective effect on neutrophil and memory T cell extravasation	[30,55]
ICAM-1	IL-6, TNF-α, IL-1β, IFN-γ	Selective effect on T cell and monocyte extravasation, deregulated in tumor EC	[31,32,55,56,57]
ICAM-2	IL-6, TNF-α, IL-1β, IFN-γ	General effect on IC extravasation	[31,32,55,56,57]
VCAM-1	IL-1β, TNF-α	Selective effect on T cell and monocyte extravasation, deregulated in tumor EC	[32,58]
JAMA	IL-1β, TNF-α, IFN-γ, IL-22, IL-17A	Selective effect on neutrophil extravasation	[10,37,38,40]
JAMC	Inflammation	General effect on IC transmigration, monocyte transmigration	[39,59]
ESAM	Inflammation	Selective effect on neutrophil transmigration	[35]
VE-cadherin (expression and phosphorylation)	VEGFA, inflammation	General effects on IC transmigration, selective effect on T cell transmigration, increased leakage	[8,9,10,23]
CD31/PECAM-1	IL-1β	Selective effect on neutrophil extravasation	[60]

## 4. Cytokines/Chemokines and Leukocyte Extravasation in Tumor Biology

The tumor immune response is highly dependent on the local cytokine/chemokine profile and these mediators can be produced by the tumor itself, IC, EC, pericytes and other stroma cells. The cytokines/chemokines may operate directly on IC, recruiting and expanding distinct populations, but also on EC changing their cell surface adhesion molecule and junctional properties. Consequently, such changes may in concert allow selective IC recruitment to the tumor.

EC P- and E-selectin are induced by tumor necrosis factor-alpha (TNF-α), interleukin (IL)-1 beta (IL-1β), IL-4 and interferon-gamma (IFN-γ) [55], ICAM-1 and -2 are induced by the cytokines IL-6, TNF-α, IL-1β and IFN-γ [55,56,57], whereas IL-1β and TNF-α induce EC VCAM-1 expression [58]. Such responses to cytokine stimulation will promote IC extravasation by facilitating leukocyte capture and crawling. Specifically, local IL-6 production may selectively promote transmigration of cytotoxic T cells by a P/E-selectin and ICAM-1-dependent mechanism(s) due to increased EC ICAM-1 expression [56]. EC JAMA is redistributed or induced by IL-1β, TNF-α or IFN-γ [10,37], thus promoting leukocyte guidance to extravasation sites at paracellular junctions. In addition, JAMA tyrosine phosphorylation is stimulated by TNF-α, IFN-γ, IL-22 or IL-17A [40], occurring in parallel with increased vascular permeability. CD31/PECAM-1 is upregulated by TNF-α, IL-1β and IFN-γ [33]. All effects of cytokines on EC cell surface expression of cell adhesion and junctional proteins will impact IC transmigration, altering the tumor immune response. Selectivity has been described for IL-1β-dependent transmigration with respect to JAMA [61] and CD31/PECAM1 dependency [60], relative to other inflammatory agents. These findings have been summarized in Table 1.

Other mechanisms besides expression of cell surface adhesion molecules and junctional proteins assist in promoting IC extravasation. Local Semaphorin3A production may help recruit tumor-associated macrophages, increase angiogenesis and cause recruitment of immunosuppressive IC [62]. IL-1β may increase EC IL-8 and monocyte chemotactic protein-1 (MCP-1)/C-C motif chemokine ligand 2 (CCL2) production, serving as a chemoattractant for neutrophils [63]. The IC profile may additionally be regulated by cell surface expression of FasL on EC as a consequence of VEGFA expression that will selectively kill CD8a-positive T cells [64]. EC cytokine production may not only affect IC extravasation but also IC functional properties. Glioblastoma EC produce large amounts of IL-6 that influence the properties of perivascular macrophages, creating a more pro-tumoral microenvironment [65] and dendritic cell maturation requires EC stem cell factor production [66].

## 5. EC-Produced Immune Checkpoint Inhibitors

Immune checkpoint proteins such as programmed cell death protein 1 (PD-1), programmed cell death ligand 1 (PD-L1) and cytotoxic T-lymphocyte-associated protein 4 (CTLA-4) are important regulators of the immune response and treatment with inhibitors of these has been proven effective in treatment of certain cancers [5]. Inflammation increases EC PD-L1 expression [67,68,69] with consequences for the tumor immune response, thus suppressing the T cell response. The efficacy of immune checkpoint inhibition is commonly increased by simultaneous anti-angiogenic treatment [68,70]. Furthermore, tumors may exploit the ability of EC to express PD-L1 for immune evasion since tumor galectin-1 production increases EC PD-L1 expression, thus reducing tumor T cell infiltration [71].

## 6. High Endothelial Venules (HEV) and Leukocyte Extravasation in Tumor Biology

Specialized post-capillary venules (HEV) comprise important sites for IC extravasation [32,72] and their properties recently have been described in an extensive review [73]. These selectively overexpress JAMB and JAMC and are particularly important for lymphocyte transmigration into lymph nodes [10,37]. Tumors may cause HEV regression, thus reducing IC tumor infiltration and the anti-tumoral immune response [74]. HEV are entry sites for IC that generate tertiary lymphoid structures which participate in immune suppressive responses [75]. In human tumors, HEV are commonly observed and in many cases associated with tertiary lymphoid structures [73]. The presence of HEV in tumors has commonly prognostic value. Data suggest that CD8^+^ cells promote HEV formation whereas Tregs will cause HEV regression [73]. Lymphotoxin-beta receptor (LTβR) stimulation is thought to induce HEV formation and consequently interference with LTβ signaling will influence HEV and immune responses [76]. Despite these findings, the underlying molecular mechanisms responsible for expansion or regression of HEV as well as their IC-recruiting properties remain poorly understood. However, combined anti-angiogenic and anti-immune checkpoint inhibition treatment will commonly increase the presence of HEV [77].

## 7. Hypoxia and Metabolites in Tumor Biology

Anti-angiogenic treatment commonly influences the tumor immune response [78], but it is difficult in most cases to distinguish between effects of the angiogenesis inhibitors on EC or direct effects of anti-angiogenic compounds on IC. Altered EC function will have an impact on local hypoxia and metabolite accumulation [15,79,80]. The tumor vasculature may or may not be adequate to supply sufficient amounts of oxygen and anti-angiogenic treatment causing vascular normalization [20] has the potential to both relieve or aggravate hypoxia, depending on the specific relevant conditions operating in that particular environment [79]. A list of publications demonstrating altered immune responses as a consequence of anti-angiogenic treatment has been provided [81]. Local hypoxia and production of metabolites may influence tumor cells, EC and IC [15,82] to secrete cytokines/chemokines that will alter the immune response, which is commonly skewed towards a pro-tumoral state with more Tregs, MDSC and type 2 macrophages [79].

Examples of tumor hypoxia that will promote a pro-tumoral immune response can be found in ovarian cancer, in which CCL28 production is increased, stimulating the recruitment of Tregs [83]. Semaphorin 3A is also increased (together with VEGFA) in hypoxic conditions in mouse Lewis lung carcinomas and this will promote the retention of pro-tumoral macrophages in the hypoxic tumor microenvironment [62]. VEGFA produced under hypoxic conditions will recruit pro-angiogenic neutrophils [84] that may participate in tumor angiogenesis and further immune response. Hypoxia may increase EC IL-8 and MCP-1/CCL2 production, both serving as chemoattractants for myeloid cells [63]. Another example is illustrated by the observation that anti-angiogenic treatment may increase tumor lymphocyte infiltration, thus enhancing the efficacy of adoptive immunotherapy [85]. Deletion of the *Eltd1* gene coding for an orphan G-protein coupled receptor reduces vascular abnormality in gliomas and enhances the efficacy of immunotherapy [86]. EC may also produce indoleamine 2,3-dioxygenase that depletes the environment of tryptophan in response to CD40-stimulating immunotherapy, thus being immunosuppressive [87]. The EC-specific *Shb* gene knockout confers altered EC properties that result in increased hypoxia in B16F10 melanoma tumors [48]. Whereas hypoxia was not investigated in a model of breast carcinoma metastasis grown in mice with *Shb*-deficient EC, increased hypoxia is a potential explanation for the augmented recruitment of MDSC observed in that setting [45]. Similarly, Sorafenib (multi-kinase inhibitor) treatment of hepatocellular carcinomas causes hypoxia that recruits immunosuppressive cells [80], including Tregs and M2 macrophages. The hypoxia in that study was primarily thought to result from angiogenesis inhibition. The recruitment of immunosuppressive cells resulted from hypoxia-induced C-X-C ligand (CXCL12) production [80].

In summary, tumor EC function may be such that hypoxia and metabolite accumulation alters the immune response, commonly promoting a pro-tumoral state by production of cytokines/chemokines. These could have direct effects on IC or indirectly via EC exert effects that cause IC extravasation.

## 8. IC and Intra-/Perivascular Location

Although it is apparent that EC and IC may be in direct contact during IC extravasation, the question remains if these cell types also remain juxtaposed for a longer time period and whether that would have functional implications. A study demonstrated increased numbers of IC juxtaposed to EC upon combined anti-angiogenic and anti-immune checkpoint inhibition and this correlated with increased IC infiltration in the tumor [88]. Similarly, in *Shb-*deficient mice, CD8^+^ cells juxtaposed to EC decreased, an effect that correlated with increased metastasis [89]. However, in that study, the *Shb* deficiency was global and thus the effect could result from cell autonomous effects in EC, pericytes, IC or fibroblasts. Cytotoxic CD4^+^ cells juxtaposed to vessels may induce EC apoptosis [90]. These studies are observational and not mechanistic and thus the relevance of IC juxtaposed to EC remains unclear. One apparent explanation is that angiocrine factors released from EC will influence the immune response by proximity. Another option was suggested by a recent publication showing the presence of transendothelial T cells (T cells protruding through the endothelial layer with simultaneous luminal and abluminal sides) in the thymus that serve a role in antigen presentation which influences the immune response [91]. I have at present no knowledge of studies demonstrating similar mechanisms operating in tumors but this would provide an attractive mechanism on how EC could influence the immune response against tumors.

## 9. Platelets and Tumor Vasculature

Platelets gain access to tumor cells by gaps in EC lining and influence tumor progression by various means, thus they play a major role in the tumor immune response [92]. They store large amounts of VEGF, transforming growth factor-beta (TGF-β) and platelet-derived growth factor (PDGF) which have direct effects on tumor cells but also promote tumor angiogenesis [93,94,95]. Depletion of platelet PDGFB reduces tumor vessel pericyte coverage, increases tumor hypoxia and metastasis [95]. IC recruitment is stimulated by platelet CXCL12 secretion upon platelet-tumor cell interaction [96]. Platelets may shield tumor cells from immunosurveillance resulting in increased metastasis [97] and may also cause tumor microthrombi with tumor embolization as a consequence [98]. Depletion of platelets will reduce vessel density and tumor perfusion [95,99].

## 10. Extracellular Vesicles (EV)

EV are microvesicles derived from cells that contain cytoplasmic content. They are formed by inward budding of endosomes in multi-vesicular endosomes which subsequently fuse with the plasma membrane. The inwardly budded vesicles are thus released to the extracellular fluid and subsequently taken up by recipient cells [100]. EV can be produced by tumor cells, EC and IC allowing communication between these players in the tumor microenvironment. Tumor cells can release EV that act on EC in a pro-angiogenic manner [100,101]. Additionally, EV can be employed to modulate EC for therapeutic purposes [102]. Inflamed adipocytes generate EV that increase EC VCAM-1 expression that in turn increases IC adhesion [103]. This suggests the possibility that a similar scenario may be operating in the tumor microenvironment, hence implicating EV-driven effects on EC that influence IC tumor infiltration. IC are readily amenable to modulation by EV [104] and thus EC-derived EV have the potential of influencing IC function. In summary, EV provide means for EC to regulate the tumor immune response although at present data are scarce indicating specific mechanisms.

## 11. Conclusions

It is clear that EC play a major role in regulating the tumor immune response. Figure 1 illustrates most of the processes that this review describe. Further complicating the situation is the fact that there are tumor-specific components as to how this occurs. Although the IC extravasation process has been studied extensively so far, only a limited number of distinct mechanisms relating to EC expression of cell surface adhesion molecules that suggest a high degree of selectivity with respect to IC transmigration have been provided. This does not exclude the possibility that hitherto undiscovered specific paths for EC-dependent IC extravasation exist. The case for altered cytokine/chemokine production in the tumor microenvironment is currently stronger. This could be a direct effect of EC producing an altered cytokine/chemokine profile but also indirect EC-dependent effects on IC, fibroblasts, other stroma cells as well as the tumor cells themselves. EC-dependent hypoxia, the presence of HEV, EC immune checkpoint expression, platelets and EV have currently been described but one could imagine vascular leakage and flow, endothelial discontinuity, pericyte and basal membrane coverage as additional factors controlling IC infiltration into the tumor, particularly in combination with other microenvironmental factors. Consequently, a high degree of selectivity may result from a concerted effect of these numerous functions operating in the particular tumor setting.

## 12. Future Perspectives

As a whole, the subject of this review is very much underexplored and methodological difficulties may be contributing to this. Although IC extravasation can be monitored by intravital imaging in real time in vivo [105,106], this is technically challenging and hard to relate to a specific underlying factor/process considering the complexity of the tumor setting and the temporal dynamics of the processes. Nevertheless, the topic is important and further studies are much warranted to obtain detailed understanding of EC regulation of tumor IC recruitment and function. As already mentioned, these processes will be tumor specific but detailed knowledge of relevant mechanisms operating on an individual basis allows for opportunities of intervention therapy of potential benefit, examples of which are relevant angiogenesis inhibition targeting specific EC functions related to infiltration of a specific IC type, interference with leakage and altering EC cell adhesion molecule/cytokine/chemokine expression.

## Figures and Tables

**Figure 1 ijms-23-02313-f001:**
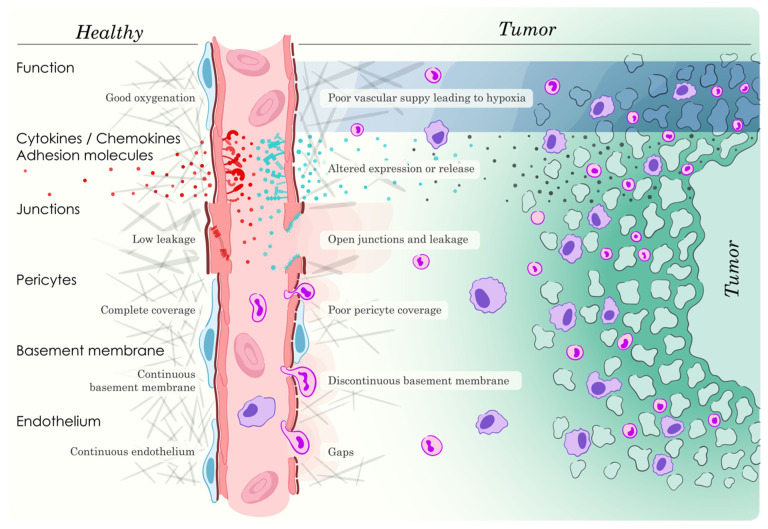
Vascular regulation of tumor IC response. Numerous processes are involved in the ability of the vasculature to influence the tumor immune response and some are illustrated in this figure. Vascular function determines the degree of tumor hypoxia, which in turn will influence cytokine/chemokine production and immune response. Oxygenation is a function of flow and oxygen diffusion, which in turn depend on vascular density and architecture. EC adhesion molecule and cytokine/chemokine expression together with the degree of junction disassembly causing open junctions will impact IC extravasation and this may be selective for certain IC or non-selective for all IC. The degree of junction disassembly will in turn depend on signaling events affecting junction protein phosphorylation and intracellular localization, focal adhesions and the cytoskeleton. Pericyte and basement membrane coverage together with gaps in EC lining are also important factors. HEV are not specifically illustrated but their abundance plays a role for tumor immune response. Eventually, the parameters mentioned will influence the recruitment of pro-tumoral and anti-tumoral IC, both adaptive and acquired, and the balance of pro-tumoral and anti-tumoral cues will decide the final immune response impacting tumor expansion or regression.

## Data Availability

Not applicable.

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
