# Peer review of "Perspectives on Vascular Regulation of Mechanisms Controlling Selective Immune Cell Function in the Tumor Immune Response"

_ijms, 2022, doi:10.3390/ijms23042313_

Round 1

Reviewer 1 Report

Dr Welsh has made a laborious work explaining the mechanisms of immune-system and vasculature crosstalk. The current review is nicely written, presenting many interesting vascular aspects that modulate the immunological response in tumor microenvironment.  Despite the fact that the review is very interesting there are some concerns that should be addressed.

  1. At the moment there are already many review articles that address the role of vasculature in the immunological responses in the tumor microenvironment (PMID: 29508855; PMID: 31497019; PMID: 31395771). This creates a concern about what new the current manuscript has to offer. It would be advisable that the author could add a subsection in all the investigated pathway addressing i. what’s new ii. where the field is going? iii. what are the advances in the field? This could justify the rationale and the novelty of the article.
  2. The current manuscript has emitted to omitted to refer to the role of platelets as platelets are key components of the blood regulating immune responses in the tumor microenvironment (PMID: 32779243). This is an element completely lacking from the manuscript.
  3. Immune checkpoints are a very hot topic in the field of immunity. Immune checkpoints are currently implicated with vascular diseases and stand as targetable molecules in oncology, upon the clinical use of Immune Checkpoint Inhibitors (PD-1inh, PD-L1inh, CTLA-4inh). Moreover, some Immune Checkpoints such as PD-L1 is expressed on the surface of the endothelial cells and can therefore be considered a vascular-dependent mechanism of immune regulation in tumors. Authors is advised to make a new and explicit section about that issue.
  4. The mechanisms explained within are very complex and the provided figure is very simplistic. Even though the artwork is nice the information provided in the review are not presented. The author is advised to generate a new figure for each section that can present in more detail the information described within.

Reviewer 2 Report

This review contains considerable information about the interactions of leukocytes with the vascular components, and how this mediates their recruitment to the tumor site. At the same time, what the review needs is a major re-structure to better clarify the topics and organize them.

First, the title can be improved and clarified; I would suggest something in line with “The role of vascular regulation during tumor immune response”. In addition, despite different immune response behaviors exist, the immune response is a unique process, and thus I would keep it in singular throughout the whole article (e.g. line 20 and 21).

Line 23 – Since the author mentioned Treg and B effectors, also Breg should be included as negative regulators of the anti-tumor immune response.

Line 92 – “Junctions and leukocyte extravasation in tumor biology”.

This is a very general title for such a long section, which include the following topics: a) A general introduction about the irregular tumor, b) angiogenesis, c) junction adhesion molecules, and  d) role of pericytes.

Structurally, the author should consider separating these sections with subtitles and a more linear order. For instance, first an overview of EC and pericytes interaction and regulation on leucocyte migration, then molecular aspects such as junction adhesion molecules and, finally, microenvironment implications (e.g. angiogenesis).

Line 136 – “down” is already to be considered as a deregulation, it can be removed.

Line 177 – Having included also cytokine and chemokine role in leukocyte extravasation, the author should also include a brief part concerning extracellular vesicles (EV). Indeed, in the last decades EV have been more and more recognized as key factors in vessel remodelling (e.g. angiogenesis) and tumor immune response regulators, thus a section dedicated on EV would fit the scope of this review.  

Author Response

Please see attachment. Response follows response to reviewer 1.

Round 2

Reviewer 1 Report

The author addressed all my comments and I believe that the manuscript is suitable for publication.

Reviewer 2 Report

The author rapidly assessed my concerns, and the revised manuscript offers an improved structure and content compared with the previous version.

I don't require further changes.